



# Comment on "Shear wave reflection seismic yields subsurface dissolution and subrosion patterns: application to the Ghor Al-Haditha sinkhole site, Dead Sea, Jordan" by Polom et al. (2018)

Michael Ezersky[1], Anatoly Legchenko[2], Lev Eppelbaum[3], Abdallah Al-Zoubi[4], Abdelrahman Abueladas[4]

[1]Geotec Engineering & Environmental Geophysics Ltd., PO Box 25031, Rishon Lezion 7502501, Israel
[2]Institute of Research for Développent, University Grenoble Alpes, Grenoble, 38 058, France
[3] Raimond and Beverly Sackler Faculty of Exact Sciences, Tel Aviv University, 6997801, Tel Aviv, Israel
[4]Engineering Faculty of Al-Balqa Applied University, Salt, 19117, Jordan

*Correspondence to*: Michael Ezersky (mikhailez@hotmail.com)

**Abstract.** We have analysed here a publication of Polom et al. (2018) on shear wave velocity reflection study in the Ghor **Al-Haditha** area (Jordan) that did not detect buried salt layer suggested earlier by other researchers. Why the modern seismic reflection method based on the S-wave technique did not detect reflections from the salt layer in the study area? The main
reason is that about ~80% of reflection lines were carried outside the salt area delineated by Ezersky et al. (2013b) based on results of El-Isa et al. (1995). Other possible factor is too strong filtering of seismic data obtained from the upper part of the section (up to 50 m deep). Our and Polom (2018) assessment of the work of other authors diverges. We affirm that the salt layer of 7-10 m thickness is located at ~40 m depth in the Ghor Al-Haditha area.

## 1 Introduction

We are a group of scientists which dealing with sinkhole appearance in the Dead Sea shores (both in Jordan and Israel) during about 20 years. We believe that the paper of Polom et al. (2018) do not quite correctly displayed the situation with geology of the Ghor Al-Haditha and analysis of the earlier published papers. Therefore, we as researchers having a rich experience in geophysical-geological analysis of sinkholes and salt layers localization in the Dead Sea shores, want to present some comments. Why the modern seismic reflection method based on the S-wave technique did not detect reflections from the salt
layer in the Ghor Al-Haditha site? Earlier we (Ezersky et al., 2017) have expressed thoughts as response to the EGU abstract (Krawczyk et al., 2015). After publication of Polom et al. (2018) we can formulate new essential arguments.





## 2 Geological context

There are only two boreholes drilled at the Ghor Al-Haditha area in 1995 (El-Isa et al., 1995) (Fig. 1, for location in plan see Fig. 2). Really, depth of the borehole 1 (eastern) was 45 m and depth of the borehole 2 (western) was 51 m. Both of them did not cross the salt layer which was suggested by Taqieddin et al. (2000). Analysing the article of Taqieddin et al. (2000) we put attention to a phrase (p. 1248 of Taqieddin et al. (2000)): "These cavities which were developed into sinkholes could be sited below a massive halite layer *presumed to exist at some 25-50 m* depth and below another one at a depth of 10-15 m. These halite layers are overlain by interbedded sand/silt and possibly clay and salt lenses or layers".

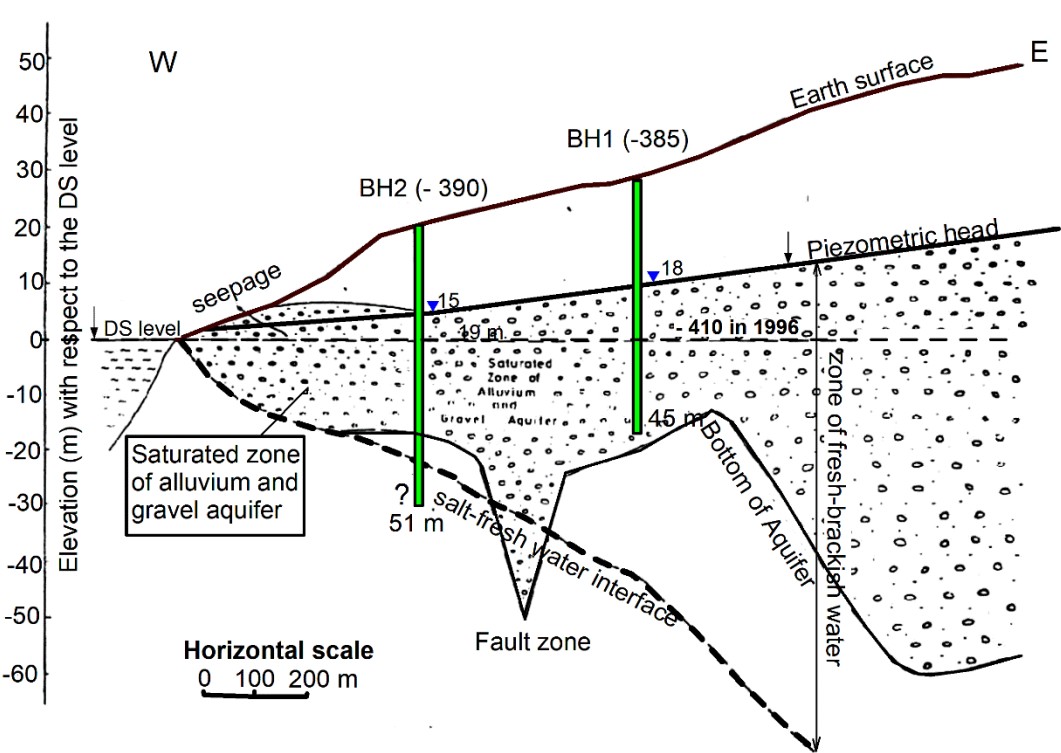

Figure 1. Hydrogeological section from El-Isa et al. (1995, Fig. 3-4, herein). The section is published for the first time

The mentioned phrase can be explained using geological and hydrogeological section from El-Isa et al. (1995) report (Fig. 1) showing that in the borehole 2 at the bottom of 10-15 m lithological data are absent. It has allowed to Taqieddin et al. (2000) to presume on a salt layer at this depth. After El-Isa et al. (1995) new boreholes were drilled by the Natural Resources Authority (NRA) of Jordan (using fresh drilling water – personal communication of A-R. Abueladas). Note that in accordance to El-Isa et al. (1995) (Fig. 1) groundwater level was established at       -405 m in borehole 2 (e.g. 30 m higher of the fresh-saline water





interface) and at      -403 m in borehole 1 (e.g. 55 m above of the fresh-saline water interface). It means that during the drilling, lower half and bottoms of both boreholes were located within the fresh or brackish groundwater. That is why we suggested that the salt cores were not extracted from the boreholes.

Polom et al. (2018) writes (p.1094, left column from above): "We also find no indication of anticlinal structures, as has to be expected from salt diapirism or salt pillows below 52 m depth". This phrase, evidently, should prove an absence of the salt. The salt diaper is located under the Lisan Peninsula (2-3 km west of the study area) (Closson, 2005). At the study (sinkhole) area the salt layer was formed in the Pleistocene–Holocene transition when catastrophic regional aridity caused intensive evaporation of DS water (Stein et al., 2010).  Therefore, this salt has a layered origin.

**3 Data acquisition**

First of all, we have considered location of the seismic reflection profiles that was carried out in second later phase of a study (Fig. 2). It was shown that many of profiles (~80%) were located out of the salt area (Fig. 2b) studied by the previous researchers (El-Isa et al., 1995; Sawarieh et al., 2000; Abueladas and Al-Zoubi, 2004; Dhemaied, 2007; Bodet et al., 2010; Frumkin et al., 2011; Ezersky et al., 2013a, 2013b) (Fig. 2a). Polom et al. (2018) show the quality of reflection raw data based

on line 1b (Fig. 5, in Polom et al.).  According to Ezersky et al. (2013b) the seismic reflection Line 1b is generally located out of the salt area. The southern part of this line is located some 200 m east of the salt area, whereas northern part is approaching to the subsidence and sinkholes area (location of this line is shown in Fig. 2b). The southern part of the line has been denoted as "stable vicinity", and raw data are characterized as of good quality. The northern part of the line is named as "strongly destabilized vicinity". Materials of this part are categorized as "poor reflection with the scattered first breaks".

Similarly, materials of the 2$^{nd}$ line acquired in the sinkhole area in 2013 were estimated as very noisy with shallow events in the single shots (Krawczyk et al., 2015). Thus, quality of the lines located west of the salt area is under doubts. However, materials of 2014 – line 2b-2 (almost completely located out of salt area) demonstrate very good quality (Fig. 6 in Krawczyk et al. (2015))

**4 Data processing**

Describing the data processing, Polom et al. (2018) write: "Because of these general improvements, some structures in the near surface down to 50 m *became weaker* than in the first iteration. This is a compromise result of combining shallow velocity results affected by wave propagation along irregular, non-straight ray paths (due to the disturbed shallow subsurface structure) with more regular straight ray path responses from deeper levels (later travel times)" (p. 1086, left column from below and right column from above). Because the salt layer location is presumed at the 35-45 m depth, this layer is in the zone of "weak

reflections" damaged by the improvement of signal during data processing.



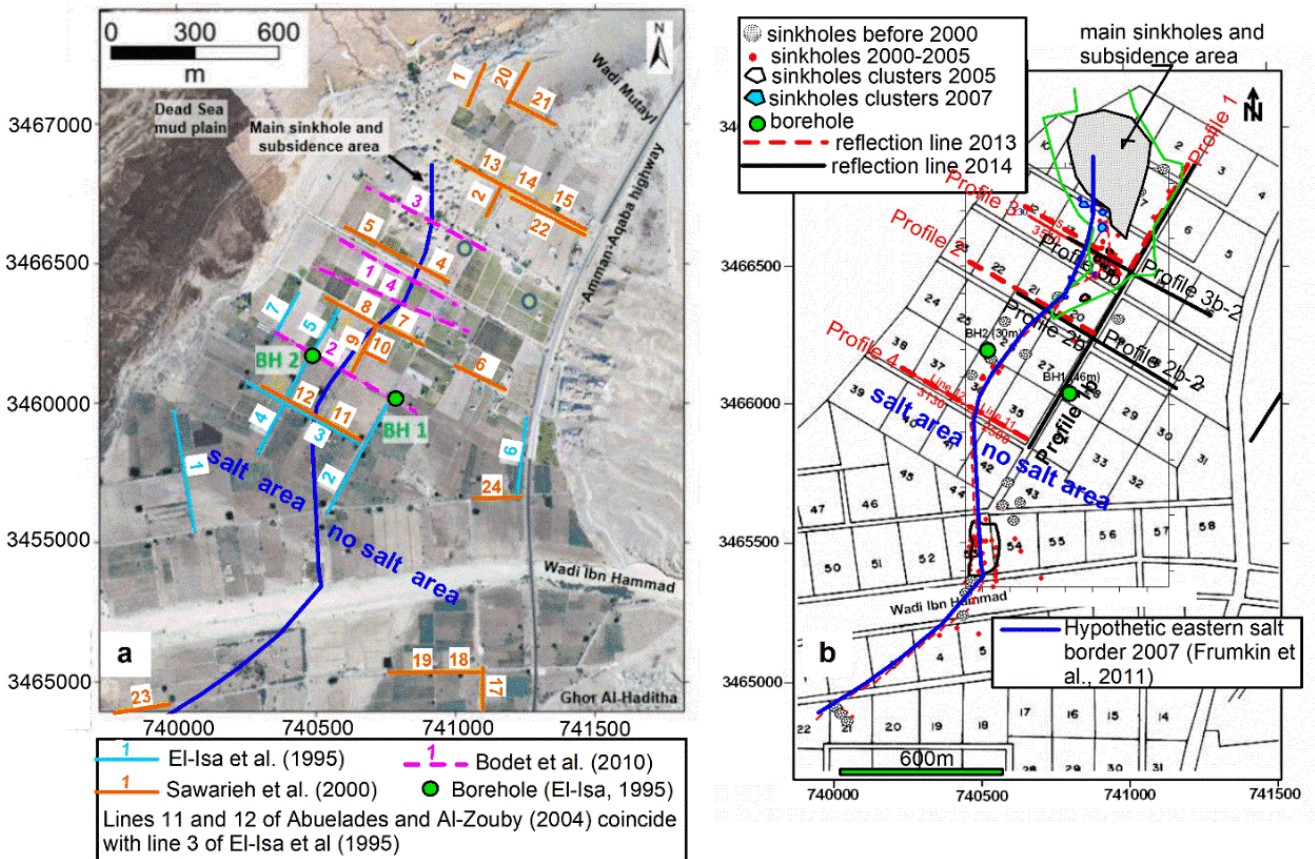

**Figure 2**. Geophysical lines layouts in the Ghor Al-Haditha area (Jordan). (a) Previous seismic line location; (b) location of the seismic reflection lines of Polom et al. (2018). Solid blue line is the eastern salt border suggested by Ezersky et al. (2013b). Map (a) is compiled by authors, map (b) is compiled on base of Fig. 4 from Polom et al. (2018).  Coordinates are in m, Universal Transverse Mercator grid.

## 5 Discussion

### 5.1. Sawarieh et al. (2000)

3. Similar confuse is with the results of Sawarieh et al. (2000). Polom et al. (2018, p.1094, right column, from above) write "In only one of 24 profiles (no. 5, Fig. 1) of Sawarieh et al. (2000), to the northwest outside the study area of El-Isa et al. (1995) and close to profile 3 of our study, Sawarieh et al. (2000) detected a P-wave velocity of 3948 m/s at 70m depth, which was interpreted as a salt diapir. Directly beside their profile 5, in their profile 4 (Fig. 1), Sawarieh et al. (2000) detected a maximum velocity of 2245 m/s at 40m depth".



If we compare locations of Sawarieh et al (2000) lines with eastern salt border in Fig. 2a one can see that the most of their

seismic refraction lines were located also out of the salt area. Only three lines (Nos. 5, 8 and 12) are located in the suggested

salt area (see Fig. 2a for location). At that, line 5 is characterized by velocity of 3948 m/s (Table 5 in Sawarieh et al.) and line

4 located in its continuation is characterized by Vp=2245 m/s. This fact testifies presence of the salt layer border between these

lines.   Line 12 is characterized by velocity of 3130 m/s (Fig. 5-18 in Sawarieh et al.).  There is some discrepancy at Line 8 in

the Sawarieh et al. (2000) study (perhaps, because of the line location inaccuracy). At other lines located outside of the salt

area, velocities less than 2500 m/s were calculated.  Note, in accordance to Ezersky (2006) velocity Vp > 2900 m/s in the Dead

Sea area characterizes a salt.

### 5.2. Thickness of salt layer

Second reason that would explain results of Polom et al. (2018) are the parameters of the salt layer calculated by authors of

the present communication using the MASW method in combination with the forward modelling (this paper of Ezersky et al.

has been submitted for publication). Derived from these calculations a depth of the salt layer varies from 37 to 42 meters from

the surface, and thickness of the salt layer is of 7-10 m. It corresponds with Polom et al. (2018) presumption on the absence of

"a thick (> 2–10m) compacted salt layer formerly suggested to lie at 35–40m depth" (p.1079 in Polom et al.) in the Ghor Al-

Haditha. Indeed, thin salt layer (2 – 10 m thick) can be transparent for seismic waves, but it does not exclude the sinkhole

hazard. Note also that in other places (p. 1096, left column from above of conclusion) Polom et al. (2018) define comparatively

thick layer that is more than 2 m thickness.

### 5.3. On salt layer possible degradation

Let us consider also a presumption of Ezersky et al. (2017) that a salt layer can be simply degraded, as it was noted by Bodet

et al. (2010).  As we have pointed above, seismic refraction and MASW investigations were carried out during 2003-2007,

whereas seismic reflection studies were conducted in late 2013-2014. This seems to be supported by hydrogeological conditions

shown in Fig. 1. Note, it is not mean that salt was completely dissolved, but it was karstified like mechanism described by

Shalev et al. (2006).

### 5.4. On applicability of seismic reflection method to mapping of unconsolidated sediments

It should be underlined that the selection of shear wave seismic reflection cannot be used alone for determination of any

lithology, especially unconsolidated sediments (Neidell, 1985; Suyama et al., 1987; Johnson and Clark, 1992). The method

cannot distinguish the unconsolidated sediments with the low Vs/Vp ratio. Yilmaz (1987) notes an ambiguity of the seismic

reflection because of the subjectivity of interpreters, selection of the different processing procedures and artefacts due to

different technique and software. Although Polom et al. (2018) note (p. 1092, right column, from below): "Due to the absence

of borehole information for depths greater than 51 m below the ground surface, more detailed interpretations remain *speculative*



and require further investigations". However, a new model of sinkhole formation is constructed basing on these results
(Krawczyk et al., 2015; Al-Halbouni et al., 2017).

## 5.5 Criticism of previous geophysical studies

We cannot agree with a criticism of Polom et al. (2018) of the previous geophysical studies (p. 1094 left column below). For instance, Park et al. (1999) suggested an empirical rule, where the normally accepted criterion of the MASW maximum penetration depth is half the wavelength maximum. However, Rix and Leipski (1991) suggested a criterion of penetration depth
of (1 to 0.5) $\lambda_{max}$ . Socco and Strobia (2004) note that, sometimes it is possible to go deeper, than half wavelength down to one wavelength (Herrmann and Al-Eqabi, 1991). Dal Moro (2017) denotes that a geophone is capable to retrieve signals at frequencies lower than its *eigenfrequency*. A rule of thumb proposes an actual "visibility limit" half the formal *eigenfrequency* (e.g., 4.5-2.25 = 2.25 Hz).

## 6 Conclusions

We have analysed here a publication of Polom et al. (2018) on shear wave velocity reflection study in the Ghor Al-Haditha area (Jordan) that did not detect buried salt layer suggested earlier by other researchers. Polom et al. (2018) have employed modern equipment and methodology. This technique is very effective and has demonstrated nice results in previous investigations of various authors. However, the salt layer was not detected in the study area because of some reasons. The main reason is that about ~80% of reflection lines were carried outside the salt area delineated by Ezersky et al. (2013b) based on
results of El-Isa et al. (1995). Other possible factor is too strong filtering of seismic data obtained from the upper part of the section (up to 50 m deep). Our and Polom (2018) assessment of the work of other authors diverges. We affirm that the salt layer of 7-10 m thickness is located at ~40 m depth in the Ghor Al-Haditha area.

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
