# Peer review of "Comment on "Shear wave reflection seismic yields subsurface dissolution and subrosion patterns: application to the Ghor Al-Haditha sinkhole site, Dead Sea, Jordan" by Polom et al. (2018)"

_Solid Earth, 2019_

## Short Comment (SC1) · 29 Oct 2019

Fig 1 is not the original as stated in the caption. It is a free modified version of the original (see scan of the original figure 3-4 of El-Isa et al. (1995) attached). A similar figure was already published by Salameh and El-Naser (2000), therein Fig 6. (Salameh, E., and El-Naser, H., The Interface Configuration of Fresh-/Dead Sea Water - Theory and Measurements, Acta Hydrochemica et Hydrobiologica 2000; 28(6); 323-328).

[Figure]

Fig 2a is not compiled by the autors as stated in the caption. It is a slightly modified copy of Fig 1 of Polom et al. (2018).

[Figure]

[Figure]

**Fig. 1.** Original fig 3-4 of El-Isa et al. 1995

---

## Author Comment (AC1) · 30 Oct 2019

**Discussion 1.**

**Polom et al. write:** Fig 1 is not the original as stated in the caption. It is a free modified version of the original (see scan of the original figure 3-4 of El-Isa et al. (1995) attached). A similar figure was already published by Salameh and El-Naser (2000), therein Fig 6. (Salameh, E., and El-Naser, H., The Interface Configuration of Fresh-/Dead Sea Water - Theory and Measurements, Acta Hydrochemica et Hydrobiologica 2000; 28(6); 323-328)

**M. Ezersky et al. response**: In the caption to Fig. 1 of the "Comment" has been stated "Hydrogeological section from El-Isa et al. (1995, Fig. 3-4, herein). *The section is published for the first time*". The presented section is slightly improved in order to specify details, captions, etc. It is not modified. One can compare the discussed sections in Fig. 1a and Fig. 1b.

Correspondingly, section published in Salameh and El-Neser (2000, Fig. 6 herein). We present this section in Fig. 1c. A caption to this figure is "*Fig. 6: Geoelectric-geologic-hydrogeologic cross-section; 1 km North of Wadi Ibn Hanunad confluence into the Dead Sea. Geoelek1rischer-geologischer-hydrogeologiscber Querschnitt, 1 Ian oordlicb der Miindung des Wadi Ibn Hammad in das Tote Meer*". Any reference to section from El-Isa et al. (1995) is absent (compare Fig. 1a and 1c). Moreover, the same section was presented in the PhD Thesis of D. Closson (2005, Fig. 15 herein) with reference to Salameh and El-Neser (2000). However, we can agree to delete a phrase *"The section is published for the first time"*.

**Thus, caption to Fig. 1 will be:** **"Figure 1. Hydrogeological section from El-Isa et al. (1995, Fig. 3-4, herein)"**

[Figure]

Figure 1. Discussion, 3 sections: (a) Fig. 3-4 From El-Isa et al. 1995,  (b) Fig. 1 from "Comment" Ezersky et al., 2019; (c) Fig. 6 from Salameh and El-Neser, 2000

**U. Polom et al. write:** Fig 2a is not compiled by the authors as stated in the caption. It is a slightly modified copy of Fig 1 of Polom et al. (2018).

**M. Ezersky et al. response**: We can agree to change this caption:

"**Figure 2. Geophysical lines layouts in the Ghor Al-Haditha area (Jordan). (a) Previous seismic line location; (b) location of the seismic 75 reflection lines of Polom et al. (2018). Solid blue line is the eastern salt border suggested by Ezersky et al. (2013b). Map (a) is slightly modified copy of Fig. 1 from Polom et al. (2018), map (b) is compiled on the base of Fig. 4 from Polom et al. (2018). Coordinates are in m, Universal Transverse Mercator grid**"

We thank Dr. U. Polom for the fruitful discussion

---

## Referee Comment (RC1) · Anonymous Referee #1 · 4 Nov 2019

Dear editors and authors, I have received a comment to a published article at Solid Earth (SE) entitled "Comment on "Shear wave reflection seismic yields subsurface dissolution and subrosion patterns: application to the Ghor Al-Haditha sinkhole site, Dead Sea, Jordan" by Polom et al. (2018)" and submitted by Michael Ezersky, Anatoly Legchenko, Lev Eppelbaum3, Abdallah Al-Zoubi, Abdelrahman Abueladas4Discussed paper published at 2018 at the same journal was entitled "Shear wave reflection seismic yields subsurface dissolution and subrosion patterns: application to the Ghor Al-

Haditha sinkhole site, Dead Sea, Jordan", authored by Ulrich Polom, Hussam Alrsh-dan, Djamil Al-Halbouni, Eoghan P. Holohan, Torsten Dahm, Ali Sawarieh and Mo-hamad Y. Atallah, and Charlotte M. Krawczyk published at Solid Earth, 9, 1079-1098. I believe that a discussion on a previous article requires evaluating both the original published article but also the submitted discussion, and during the analysis period two interactive comments have been published by the different authors. The submission of a reply article is not as usual as it is the publication of a standard article and I am less used to review this kind of articles. I believe that a reply should be done when an article represents mistakes that are known by other researchers that invalid the con-clusions from the carried out analysis in a manuscript, or when it presents an incorrect or partial data evaluation to get different conclusions than such are known from an area. In some cases when a controversy in the presented data can led to incorrect knowledge, a reply or discussion can be also of interest. Looking at the web, possibly I have just done a very short review about the rules to perform a reply paper, I have found a manuscript found at PloS comput. Biol entitlted "Ten simple rules for writing a reply paper" (PLoS Comput. Biol 2015, 11(19):e1004536. That I have followed in order to evaluate the submitted reply. Editor and authors can evaluate other rules of publishing but, at least in my opinion, it has helped in my evaluation of the submitted reply. About the original published article. Original article represents an interesting ar-ticle, with detailed descriptions of the geological context, indicating that there are some "inconsistencies" from the carried out bibliographical analysis about the distribution of the salt layer in the underground. That can be an interpretation from the carried out analysis by those authors. These authors have considered the available information for their manuscript context and they have evaluated such data and have given their interpretation about the weight that these articles represent in the geological context where they study. Moreover authors from the preliminary manuscript also evaluate previous data, in some cases unpublished, that defines the complexity of evaluation of different geophysical data in such context. From this evaluation authors consider the interest of using another approach (technique or data processing) in order to evaluate

the potential karst problems from the area and to obtain indirect information from the underground. Data description is detailed and processing explained, and problems related to the obtained results are described due to the potential characteristics of the area but also from the potential artifacts that processing can produce. This complexity was described in the introduction at the article, making reference to the potential heterogeneous series that can be present in the geological context (unconsolidated sediments) and also by the potential changes related to the karstic processes and faults in the area. Any geophysical technique has its resolution, potentiality and the possibility to obtain univocal interpretations or not depending of the available information and the consideration of the state of the art of the technique. Moreover the potential interpretation depends also from the geological context where the research has been carried out. It is obvious that indirect characterizations are not enough to obtain interpretations about properties that are different from the analyzed. In this sense, the interpretation of the geophysical records are always intuitive, open to re-evaluation depending the available information and the contour conditions defined for the interpretation and the conceptual model inferred from the expected geology from the area. Authors describe the results, and discuss data regarding different previous interpretations in the area, and potential differences to previous published articles. This seems for me a correct article that was accepted for publication, and in such case, if I have had to review it, I should suggest its publication. About the submitted reply. Ezersky et al., submit a comment to previous article highlighting some subjects about the incorrect carried out work from Polom et al., making reference to the "geological context", "data acquisition", "data processing" and about the discussion chapter from the article (thickness of salt layer, on salt layer degradation, application of seismic reflection to map unconsolidated sediments). Ezersky et al., point out a problem in the geological context evaluation from Polom et al., about the location of a salt layer in the underground that is evaluated in terms of correlation to the origin of some of the karst problems in the area. Ezersky et al., review literature from the area (both published and unpublished data) to arrive to the conclusion that part of the carried out geophysical survey in the area

has been performed in a sector where regional geology should indicate the absence of such continuous level in the underground. About data processing Ezersky et al., points out about the potential incorrect carried out data processing. While it is true that the processing must try to improve data quality to favor its interpretation, and this requires improving the visibility from shallow and deep data, in some cases this is an equilibrium that can produce that signal improvement requires a loose of resolution from the shallow data to look for changes below. In this sense, the criticism by Ezersky et al., can be correct but not means that authors have done this intentionally and moreover they have made reference to such problem in the original article. The criticism from Ezersky et al., is related with an independent potential interpretation to explain the presence of a layer of salt that can be in the area where the signal/noise ratio has decreased. However, as previously stated, the interpretation requires robust geological and direct data, and the carried out interpretation is just an independent interpretation that arise from the inverse method in geophysics. This interpretation is done without real data to be compared or to false the carried out interpretation in the preliminary article. This alternative interpretation is an interesting hypothesis, that can be correct but without data to permit to identify a mistake from the original manuscript (e.g. an independent interpretation from the same geophysical data). The rest of comments included in the manuscript make reference to what seems a trajectory of different interpretations from the same data from different researcher groups, as in some cases authors make reference to previous articles that are also discussed in the actual reply of Ezersky et al. Summary. The carried out reply points out to a mistake about the presence of a salt layer in the underground. In some cases, the state of such layer can be affected by solution decreasing the potentiality to be identified from the geophysical data. Preliminary authors pointed out to some controversy from the previous bibliographical data and evaluated the variability that can arise and the presence or absence in the underground in a sector where no direct data exists. Moreover, I am not expert in geology from the study area, but I believe that the contact of a unit, mainly related to evaporites, do not require a lonely straight contact as included at the figure 2 from the reply. The

levels can change laterally and the evaluation is being carried out from indirect data and regional correlation from something that do not outcrop. I believe that the interpretation from Ezersky et al. can be correct, but it does not decrease the interest of the original manuscript, as they pointed out the eventual controversy about such data, and the article just contextualize the geophysical data and try to interpret such data at the evaluated context. Authors' reply do not give new data or independent data that justify their interpretation, that can be correct, but they are evaluating not evident direct indicators to false what is said at the original article. They just evaluate bibliographical data that is also open to controversy as there are some different interpretations from the same matter in previous literature. Some considerations about the geophysical data and data processing are also carried out. Original authors evaluated and described them in their article. In this sense, it can be difficult to get conclusions from data that can be complex to be processed and evaluated, but preliminary authors described the identified problems producing that their interpretation are given with the required caution. All of this data produces that, in my opinion, reply article does not give any new data, it just represents that the same data can be interpreted in a different manner and it does not decrease the interest or application from the preliminary article and their robustness of their interpretations. With these data, I return to the referred article about the rules to produce a reply article that I have referred previously. I believe that there are some subjects that require a deeper analysis in order to produce a reply respect what was submitted. In this moment, is just an evaluation of other potential interpretations that goes farther than the lonely discussion of the presented data in the original article In the reply is said that authors are preparing an article about the geophysical data from the area, that is referred in the reply as Ezersky et al., at the line 94-95 "(this paper of Ezersky et al. has been submitted for publication" but is not included at the reference chapter of the manuscript and it can be the place to develop a detailed analysis with robust data and where authors can discuss their interpretations against previous one. In my opinion, the submitted reply defines a different interpretation from authors supported with previous bibliographical information that were described by Pollom et al.,

that can be interpreted as non univocal. The reply article just evaluates what someone interpreted previously that is within the resolution of the carried out approaches before in the area. The criticism about the processing and interpretation of the geophysical data is within the expected for an indirect approach, and preliminary authors describe the identified problems in the manuscript, they do not hide the identified problems, while they evaluate data with their knowledge. All of these subjects are inherent to the usual "inverse problem" in geophysics. In this sense, I believe that the submitted reply/discussion does not represent a problem with the preliminary publication, and the different interpretation that can arise from the same data can be objective of a research article where new data can be presented, where discussion can be performed and where Ezersky et al., can present their interpretation with the required supported data. In my opinion, this reply do not fit with the expected for a reply article, and in this sense, I believe that it should not be published.

---

## Author Comment (AC2) · 8 Nov 2019

Dear Anonymous Reviewer,

Thank you for your discussion denotes on our "Comments on Polom et al., 2018). Your comments are so long that to comprehensive reply to it we need to write a new article. Therefore will reply briefly.

We relate to Dr. Polom et al. with sympathy, and our comments present scientific discussion deprived any personal claims to our opponents.

At first, we have to denote that the problem of salt layer presence in the Ghor Al-Haditha area is not of academician interest only, but is very important issue for the engineering and commercial development in the Dead Sea coastal plain of Jordan. Dr. Hazim El-Nasser, Minister of Water and Irrigation and Minister of Agriculture of Jordan, said that the Red Sea – Dead Sea channel is being retarded à cause of a great number of sinkholes appeared since 1980[th] and now proliferating along the Dead Sea shore both in Israel and Jordan. All types of the industrial activity may be affected by sinkholes problem: water resources, infrastructures, agriculture, tourism, etc. The mechanism of sinkhole formation has a great practical and even political meaning. We want to involve any scientists to the discussion about the salt layer presence in the Ghor Al-Haditha (Jordan).

1. Our "Comments" does not relate to entire article of Polom et al. (2018), but only to conclusion on the salt layer absence in the study area. Moreover, we try to understand "Why the modern seismic reflection method based on the S-wave technique did not detect reflections from the salt layer in the study area?" (Abstract). We analyze some details of the study layouts (Section 3), geological interpretation of materials (section 2 and Fig. 1), some inaccuracies in interpretation of the previous researches (Subsections 5.1, and 5.5) and cite Polom et al. (2018) conclusions with respect to his interpretation of obtained results (Section 4 and Subsection 5.4).

2. You are absolutely right that "Any geophysical technique has its resolution, potentiality and the possibility to obtain univocal interpretations or not depending of the available information and the consideration of the state of the art of the technique". "Moreover, the potential interpretation depends also from the geological context where the research has been carried out. It is obvious that indirect characterizations are not enough to obtain interpretations about properties that are different from the analyzed. In this sense, the interpretation of the geophysical records is always intuitive, open to re-evaluation depending the available information and the contour conditions defined for the interpretation and conceptual model inferred from the expected geology from the area."

3. In fact, our position converges with yours about the possibility multiple interpretations of geophysical data. We only invite to write it in the conclusion of discussed paper. On the other hand, application of the method requires its

calibration. That is why, before study of salt using the Seismic Refraction method, we analyzed the geological peculiarities of the region and potential capacity of the method to reveal buried salt and to perform calibration of seismic results with the boreholes (Ezersky, 2006). It was established that in the Dead Sea graben area: (1) hard rocks apart from the salt are absent, there are only loose sediments; (2) hence, the most reliable (statistically substantiated) minimum velocity criterion for the salt layer presence (based on the P-wave velocity only) can be accepted as $Vp_{min}$=2900 m/s. Vp values less of that characterize alluvial sediments, and Vp values higher of that – characterize salt. Hereafter, values of Vp=2900 m/s and more were used as a criterion of presence of salt layer. With such a criterion, we have interpreted the seismic results of El-Isa et al. (1995).

4. Anonymous Reviewer writes: "Moreover, I am not expert in geology from the study area, but I believe that the contact of a unit, mainly related to evaporites, do not require a lonely straight contact as included at the figure 2 from the reply. The levels can change laterally and the evaluation is being carried out from indirect data and regional correlation from something that do not outcrop. I believe that the interpretation from Ezersky et al. can be correct, but it does not decrease the interest of the original manuscript, as they pointed out the eventual controversy about such data, and the article just contextualize the geophysical data and try to interpret such data at the evaluated context". You are right that the data shown in Fig. 2 do not decrease the interest to the original manuscript. We want to underline one more, the article is interesting taking into account applied modern technique and new experience. We discuss in our "Comments" the salt presence only. In this aspect, if we accept the fact of the salt boundary presence, we have to agree that the study (Polom et al., 2018) was carried out mostly out of the salt area and it can explain why authors do not see the salt layer. The model of salt formation and its relation to faults (e.g., Ezersky and Frumkin, 2013) explain shape of the salt boundary. Besides this, salt presence was suspected by other scientists (e.g., Taqieddin et al. (2000), Knight (1993)).

5. We have to agree with the Anonymous Reviewer, that "authors are preparing an article about the geophysical data from the area, that was not referred in the references". Preparing our Comments, we presumed that our above mentioned paper (submitted to another Journal) will be accepted or published before our discussion in the Solid Earth. Unfortunately, it is still under consideration and we have to change this sentence in the text. Subsection 5.2. will be rewritten so:

**"5.2. Thickness of salt layer**
Second reason that would explain results of Polom et al. (2018) are the parameters of the salt layer calculated by authors of the present communication using the MASW method in combination with the forward modelling, like described in Ezersky et al., 2013, Fig. 6, herein). Our calculations show that a depth of the salt layer varies from 37 to 42 meters from the surface, and thickness of the salt layer is of 7-10 m. It corresponds with Polom et al. (2018) presumption on the absence of "a thick (> 2–10m) compacted salt layer formerly suggested to lie at 35–40m depth" (p.1079 in Polom et al.) in the Ghor Al-Haditha. Indeed, thin salt layer (2 – 10 m thick) can be transparent for seismic waves, but it does not exclude the sinkhole hazard" (yellow marked text is new corrected edition).

6. Finally, we want to discuss with the Anonymous Reviewer his conclusion that our Comments must not be published. We believe that the Anonymous Reviewer did

not understand the goal of our comments and he will change his opinion after our reply.

**References**

Ezersky, M.G., The Geophysical Properties of the Dead Sea Salt applied to sinkhole problem. *Journal of Applied Geophysics*, 58 (1): 45-58, 2006.

Ezersky, M., and Frumkin, A., Faults—dissolution front relations and the DS sinkholes problem. *Geomorphology*, 201: 35–44, 2013. doi: 10. 1016/j.geomorph.2013.06.002.

Ezersky, M.G., Bodet, L., Akkawi, E., Al-Zoubi, A., Camerlynck, C., Dhemaied, A., and Galibert, P-Y., Seismic Surface-wave prospecting methods for sinkhole hazard assessment along the Dead Sea shoreline. *Journal of Environmental and Engineering Geophysics*, 18 (4): 233-253. doi: 10.2113/JEEG18.4.233, 2013 (Joint Issue with Near Surface Geophysics).

Knight, D.J., Extension west of Lisan Peninsula sinkholes along access road. Report on site visit 9–10 January 1993, DJK/A110/ 92235B. The Arab Potash Company, Jordan (unpublished), 1993.

Polom, U., Alrshdan, H., Al-Halbouni, D., Holohan, E.P., Dahm, T. Sawarieh, A., Atallah, M.Y. and Krawczyk, C.M., Shear wave reflection seismic yields subsurface dissolution and subrosion patterns: application to the Ghor Al-Haditha sinkhole site, Dead Sea, Jordan. Solid Earth, **9**, 1079-1098, 2018. https://doi.org/10.5194/se-9-1079-2018

Taqieddin, S.A, Abderahman, N.S., and Atallah, M., Sinkhole hazards along the eastern Dead Sea shoreline area, Jordan: a geological and geotechnical consideration. Environmental geology 39 (11): 1237-1253, 2000.

---

## Referee Comment (RC2) · Anonymous Referee #1 · 9 Nov 2019

Dear editor and authors, Thank you very much for your detailed answer and tone from the carried out answer to my previous comments to the submitted reply to Polom et al., article. As authors indicate, there are some subjects that should require a long complete manuscript to be detailed evaluated. Authors indicate that "to comprehensive reply to it we need to write a new article". This is something that I share, as the background problem can affect to security and risk from the region. In this sense, while the considered data can be used to infer a later hazard evaluation, manuscript from

Polom et al., does not evaluate this subject, or it does not evaluate that the sedimentological and paleogeographical evaluation should be change at the light of the obtained data. Polom et al., evaluate some of the available local geological data that are considered as different regarding some of the previous publications at the area (as they pointed out that they found some "inconsistencies" in the previous literature). Polom et al found that their results permitted to obtain, within the resolution of the used technique and processing methodology, an interpretation about the potential vertical series inferred from indirect data and from regional geology. These authors do not produce a conceptual hazard mapping or considerations about the security from the area, as these authors make their survey along an area where surficial karst evidences exists. This geophysical data interpretation can be evaluated later from an author or group of authors in order to evaluate other geological subjects or to perform a karst hazard analysis. However such model will be required to be congruent with both the available data and the resolution and meaning from the indirect data from the area. However, some of the considerations from the reply article comprise more about "what potentially can be done from the data" than "what the article from Polom et al., describe and analyze". I agreeing with authors in the submitted answer letter that a hazard evaluation done from these data could orient incorrectly about the real problematic in such area if this is carried out incorrectly. However this is not the objective of the manuscript and any analysis with this objective will require evaluating data in terms of resolution and potential meaning from the geophysical data. All in all, I consider that the background from the discussion is not about the geophysical model and the obtained results from the carried out processing of such data from Polom et al., and it is related with the eventual potential data that can be obtained or inferred from such data in future articles or works in the area. I believe, that in such cases, the discussion about the geophysical data can be of interest when their meaning is going to be used for other analysis. As pointed out in the letter, but also in the answer from author's reply, the background from the analysis or the discussed subjects overcame Pollom et al., article, and considering the required detailed analysis that can be needed in order to constrain hypothesis from

the area, data from the geological underground and the meaning of regional and local geological data will be needed to be discussed and evaluated. About the discussion of the propagation velocity of vp waves, they can change due to the materials and their state, considering the potential solution of part of such materials, velocity can change. At Ezersky et al., 2013, a reference is given to the expected velocity for the evaporitic unit with 2900-4080 m/s (as also included at table 2 from 2013 article). However these data, as happens with Polom et al., are obtained from the indirect evaluation of geophysical data, and they are compatible with the evaluated model for the area. However if a general evaluation is carried out for the velocity at such materials, considering that they can partially be affected by solution, they can present lower velocities than the raw salt levels. I must consider that the evaluated propagation velocity for Polom et al., it is unusually low for a salt layer, but they can present inter-beds of different materials or they can be partially karstified. In this sense, again, the same results can be interpreted in different ways. With independence of which group of authors can obtain a better approach defining the geological data from the area from their research, the arguments proposed in this moment are not balanced. There is an analysis of data from Polom et al., against some considerations pointed out by Ezersky et al. that do not false the considerations from Pollom et al. Polom et al., presented falsifiable arguments about the obtained data and, in this case, the carried out analysis at the reply's article is not refutable with the presented and given data. I invite authors to prepare a complete manuscript about this and another subjects, presenting data from the same area or a similar context in order to show their model, in a similar approach than Polom et al., carried out. This work should give an interesting article and discussion for the geological community about this interesting subject and their meaning in terms of the geological karst hazards from the study area. However, the comparison in this moment between Pollom et al., article and Ezersky et al., are not in the same scale in scientific terms. This does not mean that Ezersky et al., cannot demonstrate their arguments, it is just that they cannot be carried out in a reply as presented, and requires a complete article that is what I suggested to authors and it seems that authors are
preparing in this moment. Such article can be defined in terms of a review article in the area, where authors can present the different potential interpretations, discuss the state of the art from the used techniques and data, and then produce a discussion of all the previous available data. My evaluation is just a consideration from my personal perspective about what I consider should be a reply; however editor will evaluate the different opinions and arguments in order to get a decision. Considering that my letter was considered by the authors, and they used part of my text in order to answer, I should suggest that it is better don't to change what is said, I didn't say that the reply "must not be published", I suggested that in my opinion "it should not be published".

---

## Author Comment (AC3) · 12 Nov 2019

Dear Reviewer,

It looks like we are agreed on the technical content of our comment paper and as the paper of Polom et al. (2018) was a subject of a peer reviewing it is expected to be without technical errors. We only underline that the reported results are obtained using limited data sets and using assumptions. As well as other reported results.

The question is what should be a comment paper. We are disagreed on this subject.

If we were follow your arguments about the content of a comment paper, then it would be difficult to see where the difference between a research paper and a comment paper is.

We put this comment because Polom et al. (2018) insist that their results are obtained using modern geophysical measuring procedure and data processing. One may understand that this is the final (and assumed improved) result that replaces all previous studies. However, we have seen that it is not true and the results published by Polom et al. (2018) are within the principal uncertainty of the seismic method as well as the results published before. So, from the technical point of view, it is just another interpretation of the field measurements.

Such a conclusion is not a problem for geophysicists who have practical experience. However, it is a strong message for a wide range of non-geophysicists reading this paper who often read the summary and the conclusions. We have numerous contacts with local authorities both in Israel and in Jordan and we know that scientific results attract serious attention also of non-geophysicists.

If the Red-Dead Sea channel will be built, then it will boost the economic activity in the region and hence the subject of the natural hazard is a matter of growing importance. Local authorities, investors and decision makers need an objective expertise that is as close to the reality as possible. Unfortunately, the actual level of knowledge does not allow to know the ground truth without uncertainty.

Thus, our comment paper is a message of warning about necessary attention to pay to the reported geophysical results.

It is similar to the case when in publications it may be written "the content of this paper is under the entire responsibility of the authors and not of the journal".

With respect to your wishing that we should write the new article we note that our results and models are published worldwide and part of them devoted to Jordanian side of the Dead Sea (see references below, articles devoted to the Ghor Al-Haditha area are marked by yellow). Latter article is under consideration and we will inform you when it will be published.

**References:**

Abueladas A. and Al-Zoubi A., 2004. The application of a combined geophysical survey (GPR and seismic refraction) for mapping sinkholes in Ghor Al-Haditha Area, Jordan. Fall Meeting Supplement. EOS Transactions, American Geophysical Union, 85, 47, (Abstract GP11A-0825).

Al-Zoubi, A., Abueladas, A-R., Al-Ruzouq, R.I., Camerlynck, C., Akkawi, E., **Ezersky, M.G.,** Abu-Hamatteh, Z.S.H., Wasi, A., Al-Rawashdeh, S. 2007.  Use of 2D multi electrodes

resistivity imagining for sinkholes hazard assessment along the eastern part of the Dead Sea, Jordan. *American Journal of Environmental Sciences,* 3 (4): 229-233.

Al-Zoubi A., Eppelbaum L, Abueladas A, Ezersky M. & Akkawi E., 2013. Removing regional trends in microgravity in complex environments: Testing on 3D model and field investigations in the eastern Dead Sea coast (Jordan). International Journal of Geophysics. Special Issue. (**2013**). Article ID 341797, 13pp.
http://dx.doi.org/10.155/2013/341797

Eppelbaum L., Ezersky M., Al-Zoubi A., Goldshmidt V. & Legchenko A., 2008. Study of the factors affecting the karst volume assessment in the Dead Sea sinkhole problem using microgravity field analysis and 3D modeling. Advances in Geosciences, **18**: 1-19. www.adv-geosci.net/18/1/2008. (c) Author(s)

Ezersky, M. 2006. The Geophysical properties of the Dead Sea salt applied to the sinkhole problem. Journal of Applied Geophysics, **58**, (1): 45-58.
http://dx.doi.org/10.1016/j.jappgeo.2005.01.003

Ezersky M., 2008. *Geoelectric structure of the Ein Gedi sinkhole occurrence site at the Dead Sea shore in Israel*. Journal of Applied Geophysics, **64**: 56-69
http://dx.doi.org/: 10.1016/j/jappgeo. 12.003

Ezersky, M. and Frumkin, A., 2013. Faults—dissolution front relations and the DS sinkholes problem. Geomorphology, **201**: 35–44.
http://dx.doi.org/10.1016/j.geomorph.2013.06.002

Ezersky, M. and Livne, E., 2013. Geotechnical and geophysical properties of soils in the Dead Sea sinkhole problem. EAGE Annual Meeting of Near Surface Geoscience, 9–12 September, 2013, Bochum MO P13, 4 pp.
http://dx.doi.org/10.3997/2214-4609.20131328

Ezersky, M.G. and Goretsky, I.., 2014. Velocity-resistivity versus porosity-permeability inter-relations in Dead Sea salt samples. Engineering Geology*,* **183**: 95-115, 2014.
http://dx.doi.org/10.1016/j.enggeo.2014.09.009

Ezersky, M., and Legchenko, A., 2014. Quantitative Assessment of In-situ Salt Karstification Using Shear Wave Velocity, Dead Sea. Geomorphology, **221**: 150-163, 2014.
http://dx.doi.org/10.1016/j.geomorph.2014.06.014

Ezersky, M. and Legchenko, A., 2015. Mapping of salt consolidation and permeability using MASW method in the Dead Sea sinkhole problem. In: G. Lollino et al. (Eds.), Engineering Geology for Society and Territory, **5**: 465-469. Springer
http://dx.doi.org/10.1007/978-3-319-09048-1_89

Ezersky, M. and Frumkin, A., 2017. Evaluation and mapping of Dead Sea coastal aquifers salinity using Transient Electromagnetic (TEM) resistivity measurements. Coptes Rendus Geosciences. 349: 1-11.
http://dx.doi.org/10.1016/j.crte.2016.08.001

Ezersky M., Bruner I., Keydar S., Trachtman P. & Rybakov M., 2006b. *Integrated study of the sinkhole development site using geophysical methods at the Dead Sea western shore.* Near Surface Geophysics, **4**, 5: 335-343
http://dx.doi.org/10.3997/1873-0604.2006007

Ezersky, M., Legchenko, A., Camerlynck, C. and Al-Zoubi, A., Identification of sinkhole development mechanism based on a combined geophysical study in Nahal Hever South area (Dead Sea coast of Israel), Environmental Geology, **58**, (5): 1123-1141, 2009
http://dx.doi.org/10.1007/s00254-008-1591-7

Ezersky, M., Legchenko, A., Camerlynck, C., Al-Zoubi, A., Eppelbaum, L., Keidar S., Baucher, M., Chalikakis, K. 2010. *The Dead Sea sinkhole hazard– new findings based on a multidisciplinary geophysical study*. Zeitschrift fur Geomorph. N.F., 54, (2): 69-90, Berlin-Stutgart, http://dx.doi.org/ 10.1127/0372-8854/2010/0054S2-0069.

Ezersky M., Legchenko A., Al-Zoubi A., Levi E., Akkawi E. & Chalikakis K, 2011. *TEM study of the geoelectrical structure and groundwater salinity of the Nahal Hever sinkhole site, Dead Sea shore, Israel*. Journal of Applied Geophysics, **75**: 99-112. http://dx.doi.org 10.1016/j.jappgeo.2011.06.011

Ezersky M., Bodet L., Akkawi E., Al-Zoubi A., Camerlynck C., Dhemaied A. and Galibert P.-Y., 2013a. Seismic surface-wave prospecting methods for sinkhole hazard assessment along the Dead Sea shoreline. J. of Env. and Eng. Geoph., **18** (4), 233–253. http://dx.doi.org/10.2113/JEEG18.4.233.

Ezersky, M.G., Eppelbaum, L.V., Al-Zoubi, A., Keydar, S., Abueladas, A.-R., Akkawi, E. and Medvedev B., 2013b.Geophysical prediction and following development sinkholes in two Dead Sea areas, Israel and Jordan. Environ. Earth Sci., **70** (4), 1463-1478, http://dx.doi.org/10.1007/s12665-013-2233-2.

Ezersky, M.G., Al-Zoubi, A., Eppelbaum, L.V. and Keydar, S.,.2013c. Sinkhole Hazard Assessment of the Dead Sea area in Israel and Jordan: Multidisciplinary Study. Final Technical Report MERC_M27-050, 142pp.

Ezersky, M. G., Eppelbaum, L. V., Al-Zoubi, A., Keydar, S., Abueladas, A.-R., Akkawi, E.. and Medvedev, B., 2014. Comments to publication of D. Closson and N. Abu Karaki 1365 "Sinkhole hazards prediction at Ghor Al Haditha, Dead Sea, Jordan: "Salt Edge" and "Tectonic" models contribution - a rebuttal to "Geophysical prediction and following development ...". Environ. Earth Sci. 71 (4), 1989–1993.

Ezersky, M. G., Legchenko, A., Eppelbaum L.V., Al-Zoubi, A. 2017. Overview of the geophysical studies in the Dead Sea coastal area related to evaporite karst and recent sinkhole development. International Journal of Speleology, **46** (2), 277-302, Tampa, FL (USA) ISSN 0392-6672,https://doi.org/10.5038/1827-806X.46.2.2087

Ezersky, M., and Frumkin, A. 2017. Evaluation and mapping of Dead Sea coastal aquifers salinity using Transient Electromagnetic (TEM) resistivity measurements. C. R. Geoscience 349 (2017) 1–11 , http://dx.doi.org/10.1016/j.crte.2016.08.001

Frumkin, A., Ezersky, M., Al-Zoubi, A., Akkawi, E. and Abueladas, A.-R. 2011. The Dead Sea hazard: geophysical assessment of salt dissolution and collapse. Geomorphology, **134**, 102-117. http://dx.doi.org/10.1016/j.geomorph.2011.04.023

Legchenko A., Ezersky M., Girard J-F., Baltassat J-M., Camerlynck C. and Al-Zoubi A., 2008a. Interpretation of the MRS measurements in rocks with high electrical conductivity. Jour. of Applied Geophysics, **66**, 118-127, http://dx.doi.org/10.1016/j.jappgeo.2008.04.002

Legchenko A., Ezersky M., Camerlynck C. Al-Zoubi A., Chalikakis K. and Girard J-F., 2008b. Locating water-filled karst caverns and estimating their volume using magnetic resonance soundings. Geophysics, **73** (5):51-61. http://dx.doi.org/10.1190/1.2958007

Legchenko, A., M. Ezersky, M. Boucher, M., Camerlynck, C., A. Al-Zoubi, and K. Chalikakis, 2008c Pre-existing caverns in salt formations could be the major cause of sinkhole hazards

along the coast of the Dead Sea. *Geophys. Res. Lett*., 35, L19404, 2008c., http://dx.doi.org/ doi:10.1029/2008GL035510.

Legchenko A., Ezersky M., Kamerlynck C., Al-Zoubi A. & Chalikakis K., 2009, *Joint use of TEM and MRS method in complex geological setting*. Comptes Rendus (C.R.) Geosciences, **341**: 908-917.

Polom, U., Alrshdan, H., Al-Halbouni, D., Holohan, E.P., Dahm, T. Sawarieh, A., Atallah, M.Y. and Krawczyk, C.M., Shear wave reflection seismic yields subsurface dissolution and subrosion patterns: application to the Ghor Al-Haditha sinkhole site, Dead Sea, Jordan. Solid Earth, **9**, 1079-1098, 2018. https://doi.org/ 10.5194/se-9-1079-2018

---

## Referee Comment (RC3) · Anonymous Referee #2 · 27 Nov 2019

Dear Authors, After reading all the material included in the discussion as well as Polom et al. (2018), Ezersky et al. (2013) and Al-Halbouni et al. (2017), I think your Comment Paper has some weaknesses that prevent its publication as it is. These are points that can be addressed in your future publication and that I am sure it will make a valuable contribution to all studies related to sinkholes and subsidence in the area as your work has already shown. I think a comment paper on geophysics must show arguments about why the original paper has failures in acquisition, processing or interpretation.

[Figure]

Polom et al. (2018) summarizes the two main interpretations about the origin of the sinkholes and subsidence in the Ghor Al-Haditha region (Jordan): a salt layer below alluvial deposits (hypothesis that is supported by your group) and erosion of weak material (evaporites included within fluvial sediments) combining chemical and mechanical erosion which is used for interpretation in Polom et al. (2018). In my point of view, these are two interpretations that are not necessarily incompatible. Your comment on Polom et al. 2018 relies on 1. the assumption that the salt layer is there and that part of the profiles do not cover the area delineated as "salt layer" 2. Problems during seismic reflection processing. 3. Previous geophysical surveys. Regarding the main points introduced in your comment. a) Introduction. You contend that geological content in Polom et al. 2018 is not "quite correctly displayed". I think that they present a complete review of the geological knowledge of the area up to now. b) Introduction. You say that new essential arguments are formulated. In my opinion, I think your disagreement with the geological interpretation of Polom et al. 2018 is based on preconceived notions not in new essential arguments that I am sure that you will include in your new work. Unfortunately, I cannot see them in this comment. c) Geological context. I find this section a little bit vague. Surely, ground-truthing is a requirement for this area in order to constrain the geological interpretations. Of course, this is not always possible. d) Data acquisition. Scattering observed in seismic data is interpreted as related to the salt area. This could be more related to near-surface heterogeneities. e) Data processing. One of your hypothesis is that over- filtering has removed an expected high amplitude character of the reflection originated at the salt layer. I think that that would have affected all the reflections. I presume that a priori strong reflection from the salt should keep higher amplitude after filtering than the reflections coming from seismic contrasts within the sediments. f) Discussion. Point 5.1. This is not a new essential argument but explain results from previous work. g) Discussion. Point 5.2. I think resolution of the seismic sections shown in Polom et al. (2018) is well explained. h) Discussion. Point 5.3. You add here an interesting point that highlights monitoring as a requirement to increase knowledge of the sinkhole processes. This also has been

highlighted in Polom et al. conclusions. i) Discussion. Point 5.4. You are right that without boreholes any interpretation can be speculative but this works for seismic reflection and for any geophysical method. Anyhow, the new model of sinkhole formation is not only based on seismic but also in Al-Halbouni et al. (2017) interpretation. I do not think that this means that the salt layer model is wrong. I think that the Ghor al-Haditha zone is very peculiar since there is not any borehole information that confirms the salt layer presence. As far as I know, that layer was detected by drilling in the Israeli side. That makes interpretation of sinkhole phenomena on the Jordan side completely open. j) Discussion. Section 5.5. You are right that maximum investigation depth for surface wave methods is quite controversial. But I do not understand your point about the extension of 4.5 Hz natural frequency to a lower end (which of course is true, you have lower amplitude but you still can detect energy at lower frequencies). However, I do not think the problem for active seismic is a matter of the receiver frequency. It is more related to the source characteristics. A hammer can have problems to generate enough energy at the lower frequency. In addition, Ezersky et al. (2013) explained how maximum investigation depth was increased introducing higher modes in the inversion process. Hence, I do not think maximum depth is related to frequency anyway. I guess the modelling of fundamental and higher modes can be the best argument to support the detection of high velocity at depth. k) Last sentence of the conclusion. I think that the points introduced in the comment paper are not enough to arrive at that conclusion. In summary, Polom et al. discussion is well established and defended by data quality, processing, and uncertainties assessment. I do not see the point of publishing a comment on that paper without more datasets or ground-truthing only relying on another interpretation that of course can also be valid. I think interpretation of Polom arrives as far as possible always supported by their geophysical results. I am sure that your work will do the same with new datasets and this will be fruitful for scientific discussion since different points of view are one of the foundations for knowledge increasing.

References: Al-Halbouni, D., Holohan, E. P., Saberi, L., Alrshdan, H., Sawarieh, A., Closson, D., ... & Dahm, T. (2017). Sinkholes, subsidence and subrosion on the

eastern shore of the Dead Sea as revealed by a close-range photogrammetric survey. Geomorphology, 285, 305-324. Ezersky, M. G., Bodet, L., Akawwi, E., Al-Zoubi, A. S., Camerlynck, C., Dhemaied, A., & Galibert, P. Y. (2013). Seismic surface-wave prospecting methods for sinkhole hazard assessment along the Dead Sea shoreline. Journal of Environmental and Engineering Geophysics, 18(4), 233-253. Polom, U., Alrshdan, H., Al-Halbouni, D., Holohan, E. P., Dahm, T., Sawarieh, A., ... & Krawczyk, C. M. (2018). Shear wave reflection seismic yields subsurface dissolution and subrosion patterns: application to the Ghor Al-Haditha sinkhole site, Dead Sea, Jordan. Solid Earth, 9(5), 1079-1098.

---

## Author Comment (AC4) · 4 Dec 2019

Reply (No 4) to Reviewer #2

Dear Reviewer #2.

A comment paper is not a peer review of a published paper, but it expresses additional points on the subjects discussed in the principal paper. A comment paper may content errors or to be unacceptable for the technical reasons. It is why a comment paper is open for discussion before publishing. As far as we understand, it is not this case. In your "manuscript evaluation" message, you do not show that our comment paper is incorrect but recommend blocking it by some incomprehensible reasons. We think your position is not correct, and we do not understand why you are so against our comments. These comments do not attack the principal paper but just extend the subject discussed in.

However, we will respond your main objections.

Our comment put attention on main conclusions of the Polom's et al. (2018) paper. In the abstract Polom et al. (2018) state: "…this study aimed to clarify the subsurface characteristics responsible for sinkhole development… The most surprising result of the survey *is the absence of evidence of a thick (> 2–10 m) compacted salt layer formerly suggested to lie at ca. 35–40m dept*h. Instead, seismic reflection amplitudes and velocities image with good continuity a complex interlocking of alluvial fan deposits and lacustrine sediments of the Dead Sea between 0 and 200m depth".

The questions arise: 1. Is it geophysical paper or geological, where absence of salt is goal of this paper to insert new (really, very old) mechanism? It is exactly the same as abstract of Sc. Krawczyk (2015). 2. Is the tracing of alluvial fan deposits goal and possibility of seismic reflection method?

You write that Polom et al. (2018) present a complete review of the geological knowledge of the area up to now. You mean report published by El-Isa et al. (1995). But they do not discuss results published by Taqieddin et al. (2000) published in the respective International Journal. They mention only that Taqieddin et al. (2000) presumed a massive salt layer. Would you please explain why S. Taqieddin and M. Abdallah, participated in the El-Isa et al. (1995) survey, constructed geological model of the massive salt layer that is not corresponds to the drilled boreholes. This disagreement in interpretation can be explained only by absence of the data from the bottom of the borehole BH 2. It is what we try to explain in our Comment in Fig. 1. (By the way, only Taqieddin et al. (2000) mentioned "massive" salt layer. They do not define what means here the term "massive").

With respect to your comment (e). We present the paragraph of the Polom et al. (2018) (see line 65-70 of our initial Comments) where authors explain the result, that "Because of these general improvements, some structures in the near surface down to 50 m *became weaker* than in the first iteration". If Polom et al. (2018) write that his strong filtering has softened reflections within 50 m depth interval, how he can see salt layer at 35-40m? According to our knowledge, the data processing in seismic reflection allows to define the depth interval where filtering can be applied without influence to more deep structures.

We do not understand your sentence on Sawarieh et al. (2000). Polom et al. (2018) use results of Sawarieh as support of the salt absence concept. Any geological (geophysical) model should explain available data. If you suggest that the salt border occurs along sinkhole line, all results of the previous researchers become clearer. If we place lines of Sawarieh et al. (2000) on a map with salt border, (Fig. 2a of our initial Comments) one can see that Vp from the both sites are essentially different (Fig.5-18 below) from Sawarieh et al. 2000. Left part of the section (eastern) is characterized as low velocity (Vp < 3000 m/s) whereas right part (western is characterized as high velocity section (Vp > 3000 m/s). Border between these sections divides salt and no salt areas. The same idea is supported by Abueladas and Al-Zoubi (2004).

[Figure]

FIG.(5-18) Depth section along Profile 11+Profile 12.

And it relates directly to the salt layer concept. East to this border numerous sinkholes are formed. Similarly, line 4 and 5 are mentioned in Sawarieh et al. (2000). Most of the lines are located out of the salt area and characterized by Vp = 2200-2500 m/s.

In subsection 5.4. we are discussing on applicability of the seismic reflection method to mapping of the unconsolidated sediments, but not to the salt layer presence. You constantly try to lead us from the main problem under study to other side. However, Al-Halbouni et al. (2017) results do not give any support to reflection seismic ones. These results consider surface data that do not relates to underground structure of the subsurface. Vice versa, Al-Halbouni et al. (2017) refer to results of Polom et al. (2018) based his model on salt layer absence (subsection 5.4 of Al-Halbouni et al. (2017) (lines 1050-1070, before subsection 5.5)

You are right in paragraph J (subsection 5.4). Generally speaking, the maximum depth is related to properties of medium, minimum frequency of records, energy of the hammer, and to presence of higher modes. However, we comment conclusions of Polom et al. (2018) related to frequency of geophones and line length (p. 94, left column

(lines 29 and below)). It should be added that Bodet et al. (in Ezersky et al., 2013) used the fundamental, first and second modes and visible frequencies were lower than 4.5 Hz. So, above authors evaluated penetration depth as 60 m taking into account that in the Ghor Al-Haditha area profiling with 120m line length, 4.5 Hz geophones and 3 modes were carried out. It enabled to reach 60m deep. It is our remark.

The "Comments" is not article, but only comments to published article. Its publishing have to allow geophysical-geological and other readers be judge and do their own conclusions.

We must note that Reviewer did not disprove our arguments, and sometimes simply rejects our conclusions. The article of Polom et al. (2018) was published and we have no objection to this in any way. Authors and Reviewers of the aforementioned paper carried out a large work and our goal is not downplaying significance of this work. We want to express doubts to arguments of the authors aimed to disregard the salt layer presence in the Ghor Al-Haditha area.

In addition we want to declare our opinion that disputants should participate with their original names to exclude the possible conflicts of interests. Disputants are not reviewers of our publication. Otherwise there are interested do not allow by anywise to publish our criticism.

**References**

Abueladas, A. and Al-Zoubi, A. (2004). The application of a combined geophysical survey (GPR and seismic refraction) for mapping sinkholes in Ghor Al-Haditha Area, Jordan. Fall Meeting Supplement. *EOS Transactions*, American Geophysical Union, **85**, 47, (Abstract GP11A-0825),

Al-Halbouni, D., Holohan, E. P., Saberi, L., Alrshdan, H., Sawarieh, A., Closson, D., ... and Dahm, T. (2017). Sinkholes, subsidence and subrosion on the Interactive comment Printer-friendly version Discussion paper eastern shore of the Dead Sea as revealed by a close-range photogrammetric survey. *Geomorphology*, **285**, 305-324.

Ezersky, M.G., Bodet, L., Akawwi, E., Al-Zoubi, A.S., Camerlynck, C., Dhemaied, A. and Galibert, P.Y. (2013). Seismic surface-wave prospecting methods for sinkhole hazard assessment along the Dead Sea shoreline. *Journal of Environmental and Engineering Geophysics*, **18**(4), 233-253.

Krawczyk, C.M., Polom, U., Alrshdan, H., Al-Halbouni, D., Sawarieh, A. and Dahm, T. (2015). New process model for the Dead Sea sinkholes at Ghor Al Haditha, Jordan, derived from shear-wave reflection seismics. *Geophysical Research Abstracts*, 17, EGU2015-5761.

Polom, U., Alrshdan, H., Al-Halbouni, D., Holohan, E. P., Dahm, T., Sawarieh, A., ... and Krawczyk, C.M. (2018). Shear wave reflection seismic yields subsurface dissolution and subrosion patterns: application to the Ghor Al-Haditha sinkhole site, Dead Sea, Jordan. *Solid Earth*, **9**(5), 1079-1098.

Sawarieh, A., Abueladas, A., Al Bashish, M. and Al Seba'i, E. (2000). Sinkholes Phenomena at Ghor Al Haditha Area: Sinkholes Project (Phase 2). *Natural Resources Authority*, Internal Report No. 12, Amman, Jordan.

Taqieddin, S.A., Abderahman, N.S., and Atallah, M. (2000). Sinkhole hazard along the eastern Dead Sea shoreline area, Jordan: a geological and geotechnical consideration. *Environmental Geology*, **39**, 1237–1253.

---

## Editor Comment (EC1) · Michal Malinowski (Editor) · 11 Dec 2019

Dear authors,

After evaluation of your commentary paper, I agree with the reviewers, who indicated that your manuscript should not be published as it stands right now. Therefore I do not recommend revision but rejection at this step.

[Figure]

In my opinion, you should re-work your paper towards original research paper, showing new data and more pieces of evidence on this topic, not restricting yourself to the reinterpretation of Polom et al. work. I encourage you to consider submitting your new manuscript to Solid Earth.

Best regards, Dr Michal Malinowski SE Topical Editor